# Insights into the expanding phenotypic spectrum of inherited disorders of biogenic amines

Oya Kuseyri Hübschmann[1], Gabriella Horvath[2], Elisenda Cortès-Saladelafont[3,4], Yılmaz Yıldız[5], Mario Mastrangelo[6], Roser Pons[7], Jennifer Friedman[8], Saadet Mercimek-Andrews[9], Suet-Na Wong[10], Toni S. Pearson[11], Dimitrios I. Zafeiriou[12], Jan Kulhánek[13], Manju A. Kurian[14], Eduardo López-Laso[15], Mari Oppebøen[16], Sebile Kılavuz[17], Tessa Wassenberg[18,19], Helly Goez[20], Sabine Scholl-Bürgi[21], Francesco Porta[22], Tomáš Honzík[13], René Santer[23], Alberto Burlina[24], H. Serap Sivri[5], Vincenzo Leuzzi[6], Georg F. Hoffmann[1], Kathrin Jeltsch[1], Daniel Hübschmann[25,26,27,28], Sven F. Garbade[29], iNTD Registry Study Group*, Angeles García-Cazorla[3] & Thomas Opladen[1✉]

Inherited disorders of neurotransmitter metabolism are rare neurodevelopmental diseases presenting with movement disorders and global developmental delay. This study presents the results of the first standardized deep phenotyping approach and describes the clinical and biochemical presentation at disease onset as well as diagnostic approaches of 275 patients from the registry of the International Working Group on Neurotransmitter related Disorders. The results reveal an increased rate of prematurity, a high risk for being small for gestational age and for congenital microcephaly in some disorders. Age at diagnosis and the diagnostic delay are influenced by the diagnostic methods applied and by disease-specific symptoms. The timepoint of investigation was also a significant factor: delay to diagnosis has decreased in recent years, possibly due to novel diagnostic approaches or raised awareness. Although each disorder has a specific biochemical pattern, we observed confounding exceptions to the rule. The data provide comprehensive insights into the phenotypic spectrum of neurotransmitter disorders.

A full list of author affiliations appears at the end of the paper.

nherited disorders of neurotransmitter metabolism represent a group of rare neurometabolic diseases. They are caused by impaired biosynthesis, breakdown or transport of neurotransmitters, or of their essential cofactors, such as tetrahydrobiopterin ($BH_4$). According to the chemical structure of the primarily affected metabolite they can be classified into distinct groups[1] (Table 1):

(A) Disorders of biogenic amines (dopamine, serotonin, norepinephrine, epinephrine): (1) Primary disorders of biogenic amine metabolism: (i) Primary enzyme defects in biogenic amine biosynthesis (aromatic L-amino acid decarboxylase deficiency (AADCD), tyrosine hydroxylase deficiency (THD)); (ii) Disorders of biogenic amine catabolism (monoamine oxidase A deficiency (MAOAD), dopamine β-hydroxylase deficiency); (iii) Disorders of biogenic amine transport (vesicular monoamine transporter 2 deficiency, dopamine transporter deficiency (DATD)), (2) Disorders of tetrahydrobiopterin biosynthesis and recycling (autosomal dominant and recessive GTP-cyclohydrolase deficiency (ad/arGTPCHD), 6-pyruvoyl-tetrahydropterin synthase deficiency (PTPSD), sepiapterin reductase deficiency (SRD), dihydropteridine reductase deficiency (DHPRD), pterin-4a-carbinolamine dehydratase deficiency), (3) Co-chaperone associated disorders (DNAJC12 deficiency (DNAJC12D)) and (B) Disorders of amino acid neurotransmitters (glycine, glutamate, serine, γ-aminobutyric acid (GABA)).

Manifestations of these disorders mainly involve the central nervous system but other organ systems such as autonomic nervous, hematological or cardiovascular can also be affected. The clinical phenotype consists of a broad spectrum of symptoms, ranging from mild hypotonia and late-onset movement disorders, to early-onset lethal encephalopathies. Initial symptoms can appear at any time from the perinatal period to adulthood. Since many clinical symptoms are unspecific or overlap with features seen in other neurological conditions, such as cerebral palsy, epileptic encephalopathies and hypoxic–ischemic encephalopathy, inherited neurotransmitter disorders are often underrecognized and misdiagnosed[2]. Within these disorders only a small group can be detected via newborn screening for phenylketonuria (PKU) while other diseases require selective diagnostic tests leading to prolonged diagnostic work-up and delayed treatment initiation[3]. The outcome depends on the underlying disorder, the timing of diagnosis, initiation and type of disease-specific treatment, as well as long-term compliance to treatment[4–10].

Since inherited neurotransmitter disorders are rare disorders, the medical literature is comprised mainly of single case reports, small case series and retrospective cohort descriptions. The "International Working Group on Neurotransmitter Related Disorders (iNTD)" was founded in 2013 (www.intd-online.org), to overcome these limitations in clinical and scientific research[11]. Over the last few years it has steadily grown to include experts from 42 academic and clinical centers from 26 countries. In December 2014, iNTD set up the first international, longitudinal patient registry. This registry aims to improve our understanding of the natural history, epidemiology, genotype/phenotype correlations and clinical outcome, and to evaluate diagnostic and therapeutic strategies.

In this work, we present the first standardized evaluation of the iNTD patient registry and report comprehensive insights into pre-, peri- and postnatal presentations of inherited disorders of biogenic amines, as well as effects of initial clinical and biochemical patterns on the diagnostic process.

## Results

**Description of the study population.** Between January 1st 2015 and May 15th 2020, 429 patients were enrolled in the iNTD patient registry. Of these entries, 350 patients had a diagnosis of biogenic amine disorders. 75 patients who were transferred from the JAKE database on aromatic L-amino acid decarboxylase deficiency (http://www.biopku.org/home/jake.asp) were not analyzed in this study due to the high number of missing variables of interest. The remaining cohort of patients with disorders of biogenic amines consisted of 275 patients from 248 families (157 female (57%), from 42 countries: 196 patients born in Europe, 42 in North America, 34 in Asia, three in Central/South America and one in Africa). 109 patients had primary disorders of biogenic amine metabolism, 161 $BH_4$ deficiencies ($BH_4Ds$) and five patients DNAJC12D (Tables 2 and 3). All diagnoses were confirmed either by mutational analysis alone or by a combination of specific biochemical tests in CSF, urine and blood (Table 4).

There were no patients with dopamine β-hydroxylase deficiency or vesicular monoamine transporter 2 deficiency. For a reliable explorative analysis, a minimum number of 6 patients was required. MAOAD, DATD and DNAJC12D were included only in the descriptive analysis.

**Pregnancy, delivery and peri- and postnatal outcome.** Maternal health problems, medications taken during pregnancy and postnatal outcomes are depicted in Tables 2 and 3 and Supplementary Table 1. None of the patients were prenatally diagnosed.

There was no difference in the mode of delivery between the different primary disorders of biogenic amine metabolism (Table 2). Both in AADCD and THD a high frequency of small for gestation age (SGA) babies was noted and a remarkably high number of patients with THD had birth length (BL) < 10th percentile (Table 2 and Fig. 1).

**Table 1 Classification of inherited disorders of biogenic amines.**

**Inherited disorders of biogenic amines**

| Primary disorders of biogenic amine metabolism | | | Disorders of tetrahydrobiopterin metabolism | | Co-chaperone associated disorders |
|---|---|---|---|---|---|
| **Biosynthesis** | **Catabolism** | **Transport** | **Biosynthesis** | **Recycling** | |
| Aromatic L-amino acid decarboxylase deficiency (AADCD) | Monoamine oxidase A deficiency (MAOAD) | Vesicular monoamine transporter 2 deficiency (VMAT2D) | Autosomal dominant and recessive GTP-cyclohydrolase deficiency (ad/arGTPCHD) | Dihydropteridine reductase deficiency (DHPRD) | DNAJC12 deficiency (DNAJC12D) |
| Tyrosine hydroxylase deficiency (THD) | Dopamine β-hydroxylase deficiency (DßHD) | Dopamine transporter deficiency (DATD) | 6-pyruvoyl-tetrahydropterin synthase deficiency (PTPSD) | Pterin-4a-carbinolamine dehydratase deficiency (PCDD) | |
| | | | Sepiapterin reductase deficiency (SRD) | | |

**Table 2 Anthropometrical parameters, mode of delivery and postnatal outcomes in primary disorders of biogenic amines.**

| Disease name | Total number of patients (families) | Gestational age | | | Birth weight | | Birth length | | Birth head circumference | | | | sIUGR | Mode of delivery | Postnatal outcome in mature newborn | | Postnatal outcome in prematurity | |
|---|---|---|---|---|---|---|---|---|---|---|---|---|---|---|---|---|---|---|
| | | n | Mean | Prematurity (%) | n | <10% perc (SGA) (%) | n | <10% perc (%) | n | <10% perc (%) | | <3% perc (microcephaly) (%) | (%) | | | n | | n |
| | n | | | | | | | | | | | | | | | n | | n |
| AADCD | 56 (52) | 51 | 38.5 | 18 (n = 9) range: 32nd –36th GW | 48 | 33 | 29 | 21 | 19 | 5 | | 5 | 0 | 21 SVD, 28 C-section, 2 IVD, 2 VE, 3 NA | 28 | Feeding problems (n = 16), hyperbilirubinemia (n = 10), temperature instability (n = 4), hypoglycemia (n = 4), respiratory distress syndrome (n = 2), each once: need for mechanical ventilation, meconium aspiration, hypotonia, sleepiness, hypokinesia, tongue fasciculation, reflux, constipation, nasal congestion, tremor, jitteriness | 7 | Feeding problems, temperature instability, bradycardia, hypoglycemia, neonatal distress |
| THD | 44 (44) | 41 | 38.8 | 7 (n = 3) range: 33rd –36th GW | 37 | 49 | 23 | 39 | 16 | 31 | | 0 | 10 | 22 SVD, 17 C-section, 3 IVD, 1 VE, 1 NA | 13 | Hyperbilirubinemia (n = 5), feeding problems (n = 3), each once: need for mechanical ventilation, sepsis, hemorrhage, sleepiness, umbilical sepsis, club feet | 3 | Feeding problems, temperature instability, bradycardia, hypoglycemia, neonatal distress |
| MAOAD | 4 (3) | 4 | 40 | 0 | 4 | 0 | NA | NA | 0 | NA | | NA | NA | 2 SVD, 2 VE | 1 | Feeding problems, hypotonia, excessive sedation | NA | |
| DATD | 5 (5) | 5 | 39 | 0 | 4 | 25 | 2 | 50 | 2 | 0 | | 0 | 0 | 2 SVD, 3 C-section | 3 | Feeding problems (n = 2), hyperbilirubinemia, temperature instability, resuscitation, need for mechanical ventilation | NA | |

C-section cesarean section, GW gestational week, IVD induced vaginal delivery, n number reported, NA not available, perc percentile, SGA small for gestational age, sIUGR symmetrical intrauterine growth restriction, SVD spontaneous vaginal delivery, VE vacuum extraction.

**Table 3 Anthropometrical parameters, mode of delivery and postnatal outcomes in BH$_4$ deficiencies and DNAJC12 deficiency.**

| Disease name | Total number of patients (families) | Gestational age | | | Birth weight | | Birth length | | Birth head circumference | | | sIUGR | Mode of delivery | Postnatal outcome in mature newborn | | Postnatal outcome in prematurity | |
|---|---|---|---|---|---|---|---|---|---|---|---|---|---|---|---|---|---|
| | n | n | Mean | Prematurity (%) | n | <10% perc (SGA) (%) | n | <10% perc (%) | n | <10% perc (%) | <3% perc (microcephaly) (%) | (%) | | n | | n | |
| arGTPCHD | 18 (17) | 17 | 40 | 0 | 15 | 33 | 8 | 25 | 7 | 29 | 0 | 8 | 9 SVD, 8 C-section, 1 NA | 8 | Jitteriness (n = 2), hyperbilirubinemia (n = 2), each once: tremor, irritability, feeding problems, vomiting, temperature instability | NA | |
| adGTPCHD | 36 (25) | 33 | 39.4 | 3 (n = 1) 36th GW | 27 | 11 | 15 | 13 | 8 | 0 | 0 | 0 | 29 SVD, 5 C-section, 1 IVD,1 VE | 3 | Hypoglycemia, congenital torticollis, tachypnoea due to a small ventricular septal defect | 1 | None |
| PTPSD | 56 (53) | 49 | 38.2 | 10 (n = 5) range: 33rd-36th GW | 45 | 56 | 36 | 33 | 29 | 34 | 24 | 18 | 20 SVD, 24 C-section, 4 IVD,1 VE, 7 NA | 17 | Hyperbilirubinemia (n = 8), feeding problems (n = 6), hypoglycemia (n = 5), each once: sepsis, hypothermia, floppiness, respiratory distress | 2 | Hypoglycemia, hyperbilirubinemia, feeding problems, temperature instability |
| DHPRD | 37 (34) | 36 | 38.7 | 11 (n = 4) range: 31st-36th GW | 29 | 24 | 16 | 6 | 15 | 27 | 7 | 4 | 21 SVD, 11 C-section, 4 IVD, 1 NA | 12 | Hyperbilirubinemia (n = 6), feeding problems (n = 6), hypoglycemia (n = 2), each once: sepsis, hypoxic ischemic encephalopathy, oculogyric crisis, hypertonia, meconium aspiration, dysplastic left kidney | 3 | Neonatal distress, tachycardia, ascites |
| SRD | 14 (11) | 14 | 38.6 | 21 (n = 3) range: 33rd-36th GW | 10 | 30 | 7 | 29 | 5 | 40 | 0 | 11 | 8 SVD, 3 C-section,3 IVD | 4 | Hyperbilirubinemia (n = 2), feeding problems, perinatal hypoxia | 3 | Hyperbilirubinemia, feeding problems, need for mechanical ventilation |
| DNAJC12D | 5 (4) | 5 | 37.8 | 20 (n = 1) 36th GW | 5 | 0 | 5 | 20 | 5 | 0 | 0 | 0 | 3 SVD, 2 C-section | 2 | Hyperbilirubinemia | 1 | None |

C-section cesarean section, GW gestational week, IVD induced vaginal delivery, n number reported, NA not available, perc percentile, SGA small for gestational age, sIUGR symmetrical intrauterine growth restriction, SVD spontaneous vaginal delivery, VE vacuum extraction.

**Table 4 Age at initial symptoms and diagnosis as well as diagnostic methods.**

| Disease name | Total number of patients | Age at initial symptoms (months) | | | | Age at diagnosis (months) | | | | Diagnostic work-up | | Confirmatory methods (number of available results) |
|---|---|---|---|---|---|---|---|---|---|---|---|---|
| | n | n | mean | min | max | n | mean | min | max | Type of diagnostic screening | Results of NBS for PKU | |
| AADCD | 56 | 49 | 2 | 0 | 12 | 51 | 41.2 | 0.9 | 389.3 | 54 SS, 1 HRFs, 1 NA | NA | mutational analysis (46/56) ± biogenic amines in CSF (42/56) / AADC enzyme activity in plasma (23/56) |
| THD | 44 | 41 | 6.6 | 0 | 60 | 43 | 64.6 | 2 | 474.5 | 43 SS, 1 HRFs | NA | mutational analysis (41/44) ± biogenic amines in CSF (32/44) |
| MAOAD | 4 | 3 | 10.3 | 1 | 24 | 4 | 177.5 | 6.5 | 447 | 4 SS | NA | mutational analysis (3/4) ± biogenic amines in CSF (1/4) |
| DATD | 5 | 4 | 3 | 1 | 5 | 5 | 62.1 | 9 | 194.7 | 5 SS | NA | mutational analysis (3/5) ± biogenic amines in CSF (3/5) |
| arGTPCHD | 18 | 16 | 23 | 0 | 204 | 16 | 31 | 0.7 | 170.3 | 15 SS, 2 NBS*, 1 HRFs | 7 normal, 8 HPA (2 NBS, 6 SS), 3 not done | mutational analysis (14/18) ± biogenic amines in CSF (10/18)/pterins in CSF (8/18)/pterins in urine (7/18) |
| adGTPCHD | 36 | 31 | 80 | 6 | 516 | 35 | 155.6 | 18 | 632.7 | 30 SS, 5 HRFs, 1 NA | 23 normal, 6 not done | mutational analysis (36/36) |
| PTPSD | 56 | 33 | 10.7 | 0 | 192 | 54 | 19.6 | 0.1 | 401.5 | 15 SS, 37 NBS*, 2 HRFs, 2 NA | 43 HPA (37 NBS, 4 SS, 2 HRFs), 11 not done | mutational analysis (39/56) ± biogenic amines in CSF (25/56)/pterins in CSF (20/56)/pterins in urine (31/56) |
| DHPRD | 37 | 24 | 10.3 | 1 | 72 | 37 | 21.4 | 0.4 | 255.5 | 22 SS, 13 NBS*, 2 HRFs | 1 normal, 30 HPA (13 NBS, 17 SS), 5 not done, 1 NA | DHPR enzyme activity in blood (23/37) ± mutational analysis (24/37)/biogenic amines in CSF(18/37)/pterins in CSF (16/37)/pterins in urine (10/37) |
| SRD | 14 | 12 | 3.1 | 0 | 6 | 14 | 114.1 | 7 | 316.3 | 13 SS, 1 HRFs | 10 normal, 4 not done | mutational analysis (13/14) ± biogenic amines in CSF (9/14)/pterins in CSF (5/14) |
| DNAJC12D | 5 | 5 | 21.6 | 6 | 48 | 5 | 111.2 | 73 | 170.3 | 2 SS, 2 NBS*, 1 HRFs | 2 normal, 3 HPA | mutational analysis (5/5) |

*indicates that diagnosis was established immediately following HPA on NBS. CSF cerebrospinal fluid, HRFs high-risk family screening, n number reported, NA not available, NBS newborn screening, PKU phenylketonuria, SS selective screening.

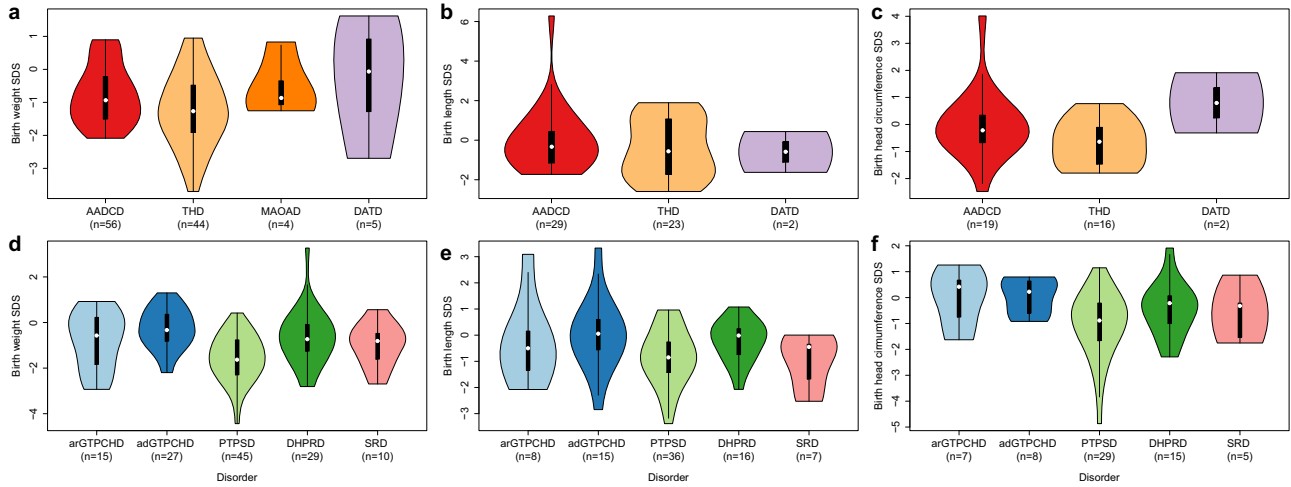

**Fig. 1 Birth metrics.** Distribution of birth metrics in primary disorders of biogenic amine metabolism (**a**–**c**) and BH deficiencies (**d**–**f**). White spot: median, black bars: interquartile range (25% and 75%), n: number reported.

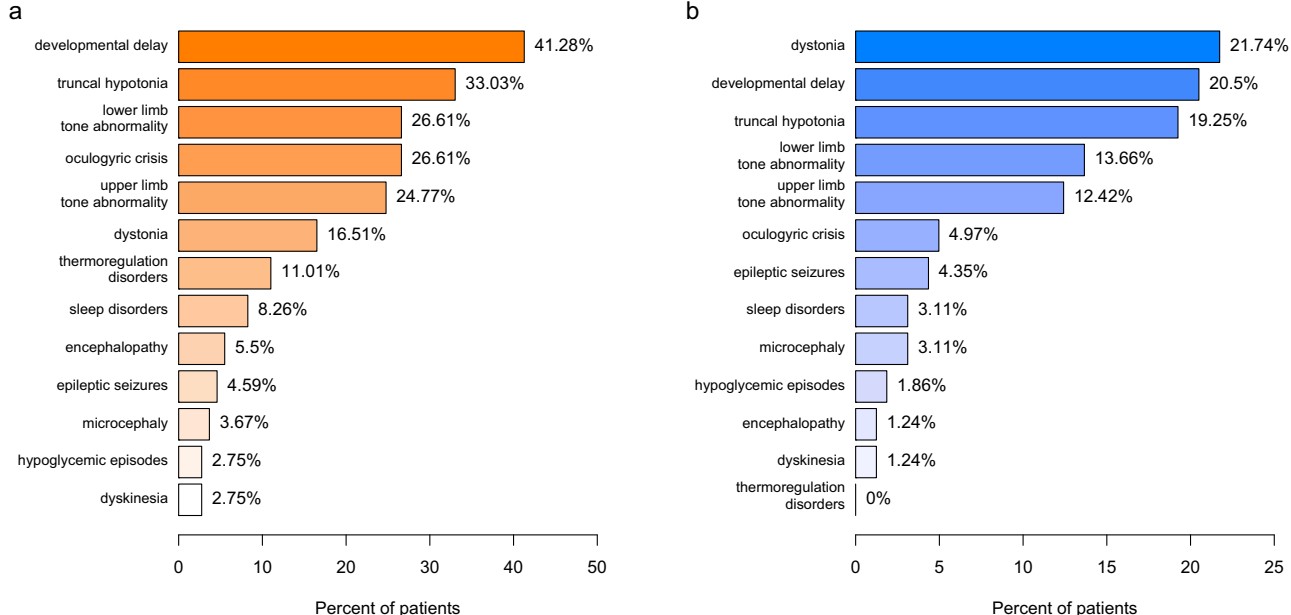

**Fig. 2 Symptoms at disease onset.** Initial symptoms in primary disorders of biogenic amine metabolism (**a**) and BH deficiencies (**b**), age independent analysis.

In BH4Ds, 27 pregnancies with maternal health issues were reported. Among four mothers affected with adGTPCHD in this group, two mothers did not require any drug treatment despite having intermittent dystonia. There was no difference with regard to the mode of delivery between arGTPCHD and PTPSD while spontaneous vaginal delivery (SVD) was the most frequent mode of delivery in adGTPCHD and DHPRD (Table 3). Interestingly, newborns with PTPSD showed the highest rate of symmetrical intrauterine growth restriction (sIUGR) and SGA (Table 3 and Fig. 1).

**Initial clinical presentation.** The most common initial symptoms in AADCD and THD were developmental delay and truncal hypotonia (Fig. 2a). Thermoregulation disorders and oculogyric crises were prominent in AADCD and occurred more often than in THD (Fig. 3a). Dystonia was similarly frequent in both disease

groups. Sleep disturbances were only observed in AADCD. The following additional symptoms (not listed in Figs. 2a and 3a) were reported in AADCD: gastrointestinal symptoms ($n = 7$, 12.5%), nasal congestion, ($n = 6$, 11%), tremor ($n = 3$, 5%) and hypokinetic rigid syndrome ($n = 3$, 5%). In THD, tremor ($n = 9$, 20.5%), hypokinetic rigid syndrome ($n = 3$, 7%), gastrointestinal symptoms ($n = 3$, 7%) and suspected seizures ($n = 3$, 7%) were reported. Developmental delay was reported in two out of four patients with MAOAD, epileptic seizures and encephalopathy in one and sleep disorders the other patient. Developmental delay, hypomimia and drooling were reported in one patient with DATD and dystonia and hypokinesia in another patient. Microcephaly and irritability each were the initial symptoms in other two patients with DATD.

The mean age at onset of first symptoms was 4.3 months across all disorders in this group (range 0–60 months, Table 4). Patients with AADCD were already symptomatic in the neonatal period

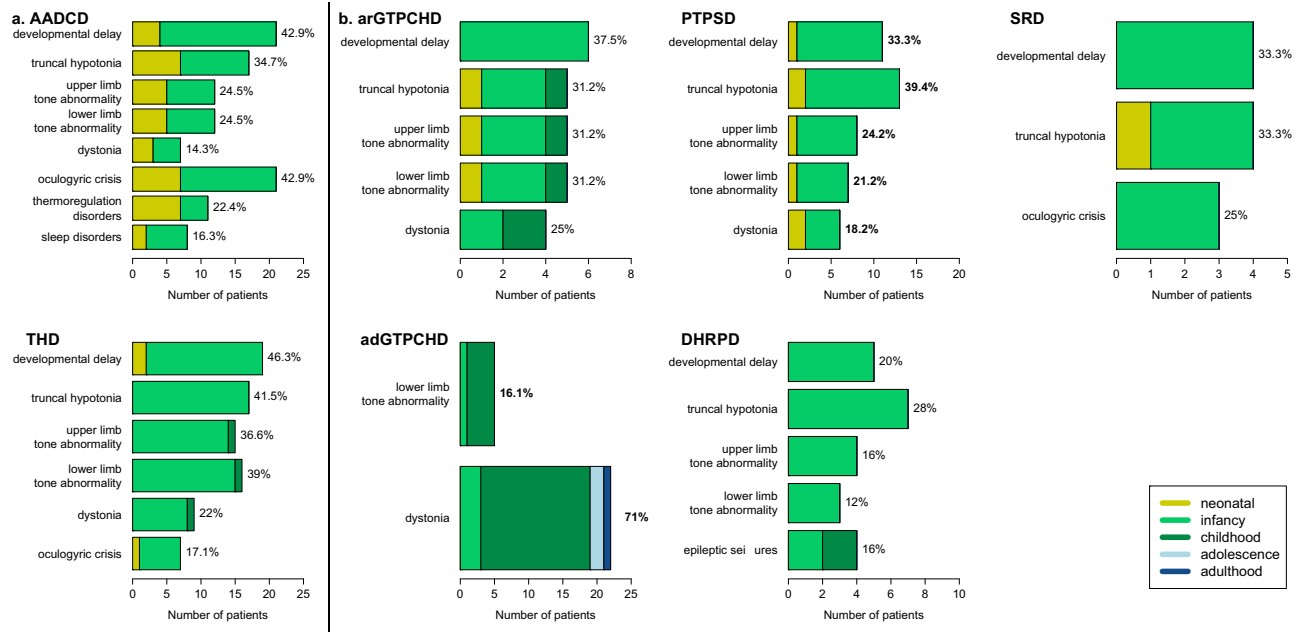

**Fig. 3 Age dependency of initial symptoms.** Age-dependent onset of initial symptoms in primary disorders of biogenic amine metabolism (**a**) and BH deficiencies (**b**). Initial symptoms reported in minimum 5 patients––except in arGTPCHD, DHPRD and SRD in minimum 3 patients. The percentages correlate with available data about both occurrence of the symptom and age at onset of disease and not to the whole disease group.

and in early infancy while the majority of patients with THD became symptomatic during infancy (between 1 and 15 months, Fig. 3a). Only three out of 33 patients with THD were symptomatic neonatally with developmental delay and oculogyric crises while two patients showed initial symptoms in childhood (aged 3 and 5 years), one with hypotonia of extremities, dyskinesia and dystonia, and the other with gait disorder and speech difficulties.

The most common initial symptoms among BH4Ds were dystonia, developmental delay and truncal hypotonia (Fig. 2b). Symptom onset occurred most frequently during the neonatal period or infancy in all patients with BH4Ds except for those with adGTPCHD (Table 4 and Fig. 3b). This neonatal/ infantile initial presentation was typically characterized by developmental delay and (truncal) hypotonia. In contrast, dystonia (61%) and lower extremity hypertonia (13%) were the most frequent initial symptoms in adGTPCHD, together with gait disorders ($n = 6$, 16%), orthopedic problems ($n = 5$, 14%), tremor ($n = 3$, 8%) and toe walking ($n = 3$, 8%, not listed in Figs. 2b and 3b). Dystonia was more frequently seen in arGTPCHD (22%) than in PTPSD (11%), DHPRD (5%) and SRD (7%) in an age-independent analysis (Fig. 3b, age-dependent analysis). Seizures were reported only in PTPSD (5%) and DHPRD (11%) among all BH4Ds. As additional initial symptoms, hypokinetic rigid syndrome was reported in PTPSD ($n = 3$, 5%) and DHPRD ($n = 3$, 8%) and failure to thrive in PTPSD ($n = 3$, 5%). Of note is the higher frequency of oculogyric crises (21%), and sleep disorders (14%) in SRD compared to other BH4Ds (0–5% and 0–6% respectively; age-independent analysis). Developmental delay alone was reported in one patient with DNAJC12D, together with oculogyric crises in another patient and behavioral problems in two patients.

**Diagnostic work-up and diagnostic delay**. The latency to diagnosis was comparably long for AADCD (mean = 41 months, range 0–386 months) and THD (mean = 45 months, range −6–361 months) (Fig. 4a). Patients with AADCD born before 2009 had a statistically significant longer latency to diagnosis

(mean 68.1 months) than those born after 2009 (mean 7.3 months, $p = 0.00041$, WMW-test) (Fig. 4b). In the group of patients with THD, this pattern was also found for patients born before and after 2005, respectively (mean diagnostic delay of 89.8 months vs. 14.4 months, $p = 0.00041$, WMW-test).

Investigating the effect of symptoms on age at diagnosis or on the diagnostic delay, we found that presentation with hypotonia, seizures, encephalopathy, microcephaly, sleep disturbances or thermoregulation disorders was associated with earlier age at diagnosis (3.2 years) and with less diagnostic delay (30 months), than presentation with dystonia, dyskinesia, hypoglycemia or developmental delay (diagnosis age 4.9 years, $t (88.14) = −1.68$; $p = 0.1$; diagnostic delay 45 months, $t (175.43) = −1.89$; $p = 0.06$; Fig. 5a and b).

Since disorders presenting with hyperphenylalaninemia (HPA), arGTPCHD, PTPSD and DHPRD, can be detected by newborn screening (NBS) for phenylketonuria, they would be expected to be diagnosed at a younger age and with less delay than those without HPA. In our study cohort, diagnostic work-up as well as diagnostic delay varied strongly depending on the occurrence of HPA (Table 4 and Fig. 4a). In patients with arGTPCHD, PTPSD and DHPRD, HPA on NBS led to significantly shorter diagnostic delay than in cases without HPA or without any NBS performed (mean 3.1 vs. 39.6 months, $t (24.05) = 2.99$; $p = 0.006$, Fig. 4c). There was no significant difference in diagnostic delay between the latter group vs. those BH4Ds without HPA and AADCD and THD (mean 39.6 vs. 76.7 vs. 51 months, $t (2) = 1.89$; $p = 0.15$).

Disorders without HPA (adGTPCHD and SRD) were mainly diagnosed by selective screening after onset of symptoms or by high-risk family screening (HRFs). In our study, the majority of patients with adGTPCHD were symptomatic in childhood and had a mean diagnostic delay of 60 months (range 1–245 months). Most patients with SRD were symptomatic in the first 6 months of life, however, this group in particular showed a prominently prolonged latency to diagnosis with an average duration of 112 months (range 12–316 months, Table 4, Fig. 4a). SRD patients born after 2009 had a significantly shorter diagnostic delay than those born before (mean 140 months, vs. 29.9 months,

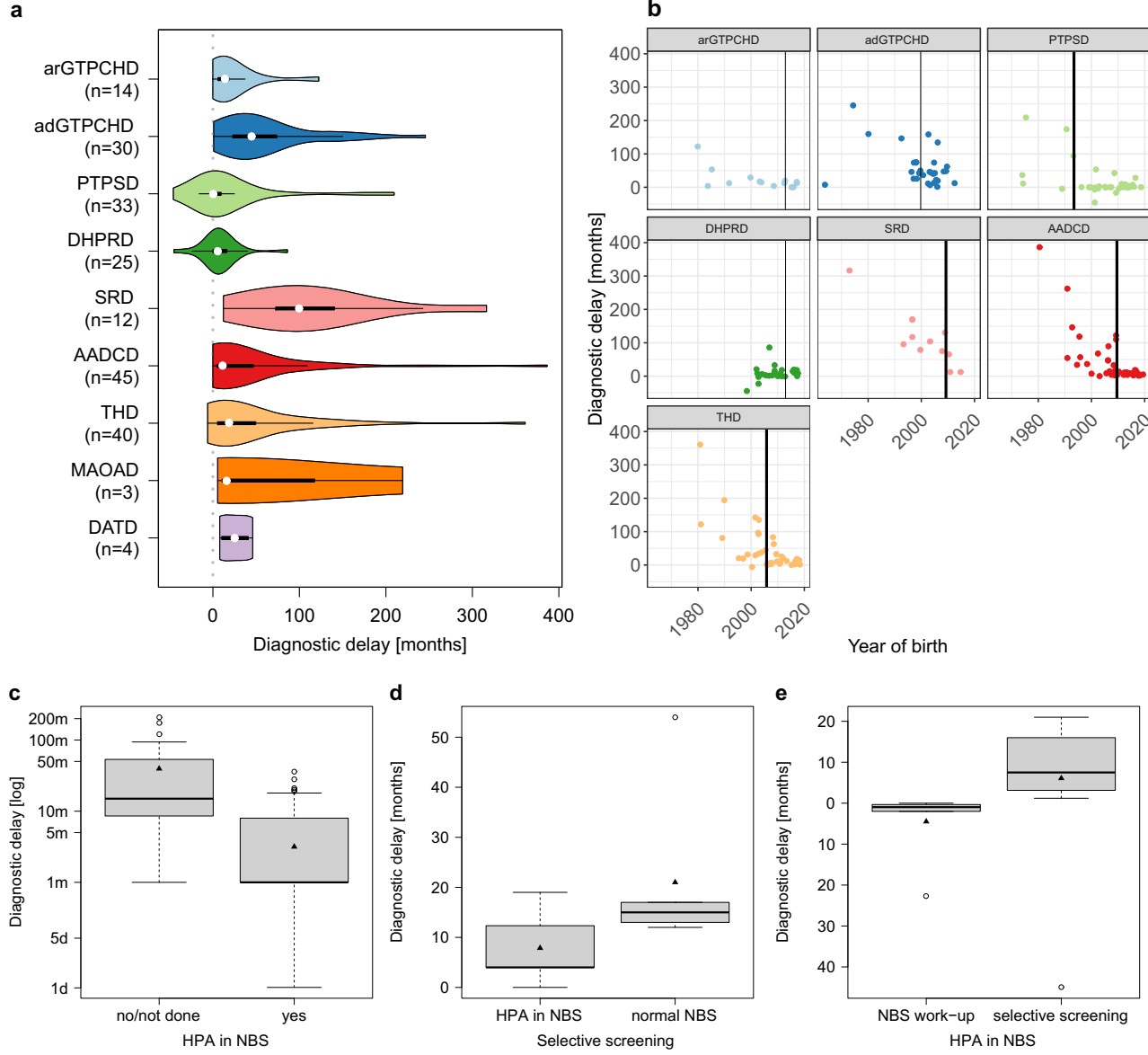

**Fig. 4 Effects on diagnostic delay in different diseases. a** Diagnostic delay in disorders of biogenic amine metabolism and BH deficiencies. White spot: median, black bars: interquartile range (25% and 75%), n: number reported. **b** Effect of date of birth on diagnostic delay; bold line: significant, two-sided WMW-test, $p < 0.05$, **c** Diagnostic delay in arGTPCHD, PTPSD and DHPRD in patients without any or normal newborn screening (NBS) results ($n = 24$) vs. patients with hyperphenylalaninemia (HPA) on NBS ($n = 47$); two-sided $t$-test, $p = 0.006$. **d** Diagnostic delay in arGTPCHD regarding HPA on NBS ($n = 5$) vs. normal NBS ($n = 6$); two-sided $t$-test, $p = 0.11$. **e** Diagnostic delay in DHPRD in patients having HPA on NBS diagnosed directly after NBS ($n = 6$) vs. via selective screening ($n = 14$); two-sided $t$-test, $p = 0.08$. Diamond: arithmetic mean, bold line: median, boxes present interquartile range (25% and 75%).

$p = 0.021$, WMW-test). For adGTPCHD no significantly discriminating date was found (Fig. 4b).

Seven patients with arGTPCHD had normal NBS results without HPA. Eight patients presented with HPA on NBS but in six of these eight cases the diagnostic work-up was initiated later (Table 4). In patients with arGTPCHD who were diagnosed via selective screening, the mean diagnostic delay in the group with HPA on NBS (7.8 months) differed from the other group with normal results on NBS (21.5 months) without reaching statistical significance ($t$ (7.39) = −1.78; $p = 0.11$, Fig. 4d).

Only in the PTPSD group did all the available NBS results demonstrate HPA (Table 4). While the vast majority of the patients with DHPRD (30/37) had HPA on NBS, one patient surprisingly had a normal NBS and presented normal

phenylalanine levels in plasma but high levels in CSF on repeated measurements. This patient was homozygous for a new variant in the *QDPR* gene (NM_000320.3, c.466 G > A, p. Ala156Thr). There were 17 DHPRD patients whose diagnosis was established late despite having HPA on NBS and who had a longer mean diagnostic delay (6 months) than those who were diagnosed by specific work-up immediately following the detection of HPA on NBS (-4.5 months, $t$ (16.44) = −1.87; $p = 0.08$, Fig. 4e). Negative values for the diagnostic delay in PTPSD and DHPRD in Fig. 4a are explained by an early diagnosis via NBS or HRFs before onset of symptoms. The lowest values for the mean diagnostic delay (8 months) and for the maximum diagnostic delay (86 months) were recorded in DHPRD.

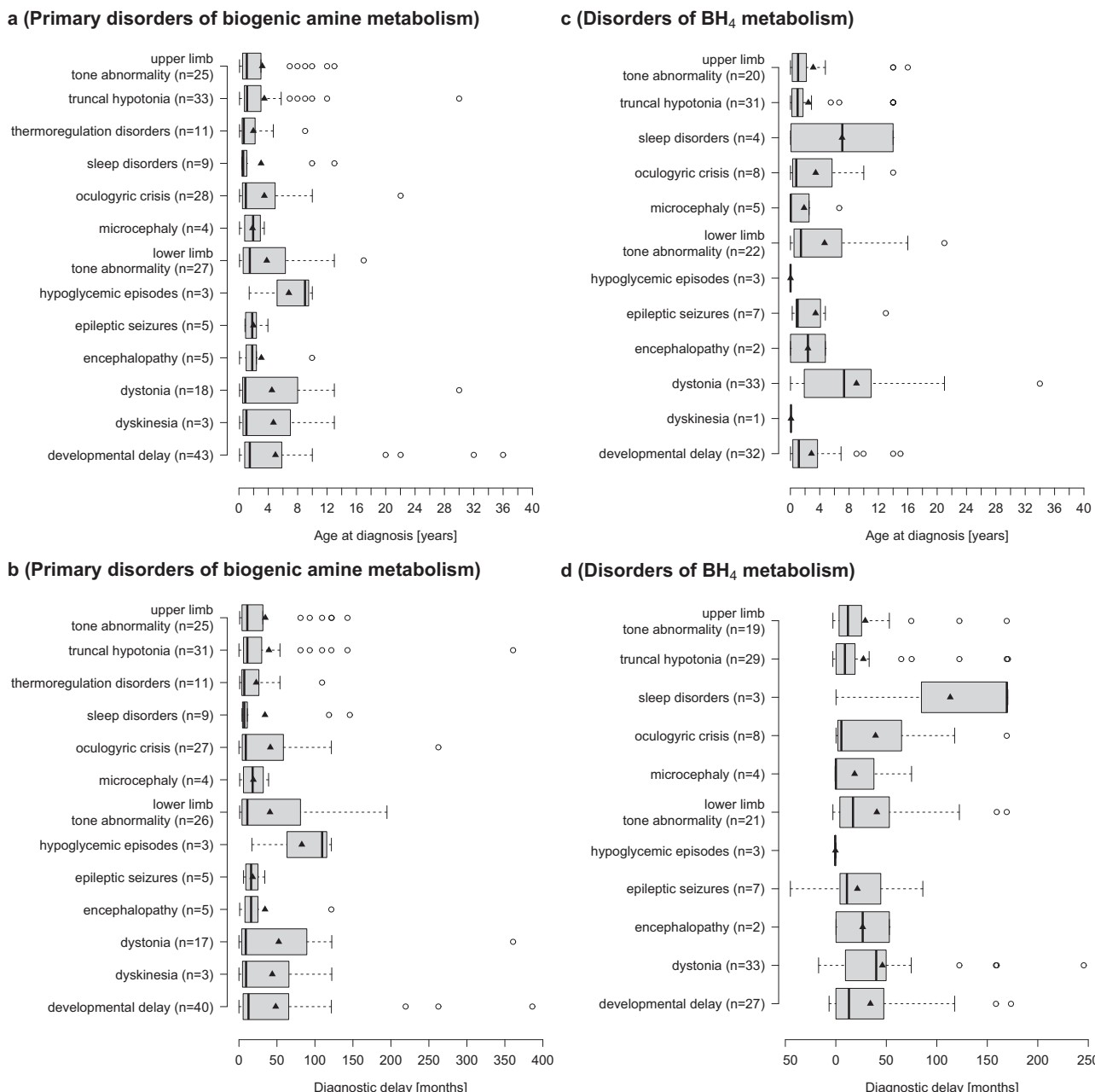

**Fig. 5 Association between initial symptoms and age at diagnosis vs. diagnostic delay.** Association between initial symptoms and age at diagnosis vs. diagnostic delay in disorders of primary biogenic amine metabolism (**a**, **b**) and BH metabolism (**c**, **d**). Diamond: arithmetic mean, bold line: median, boxes present interquartile range (25% and 75%).

We identified the birth year 1993 as the most strongly discriminating and statistically significant time point regarding changes in the latency to diagnosis for PTPSD (mean 86.6 months, SD 88.6 months vs. mean 3.4 months, SD 17.3 months, p = 0.028, WMW-test, Fig. 4b). In the remaining BH₄Ds, we could detect a trend around the years 1999 for arGTPCHD and 2012 for DHPRD but these dates did not reach statistical significance.

Truncal hypotonia, upper limb hyper-/or hypotonia, developmental delay, epilepsy, encephalopathy, microcephaly, thermoregulation disorders, oculogyric crises, dyskinesia or hypoglycemia were associated with earlier age at diagnosis (2.7 years) than lower limb hypo-/or hypertonia (4.7 years) or dystonia and sleep disorders (8.8 years; ANOVA; $F$ (2,165) = 14.89; $p = 0.16$ for 2.7 vs 4.7 years; $p = 0.0000005$ for 2.7 vs 8.8 years; $p = 0.02$ for 4.7 vs.

8.8 years). Other than in primary disorders of biogenic amine metabolism, developmental delay, tone abnormalities in upper limb and trunk as well as epilepsy were associated with a shorter diagnostic delay (28 months) than oculogyric crises, dystonia, lower limb tone abnormalities and sleep problems (47 months, $t$ (117.73) = −2.15; $p = 0.03$, Fig. 5c, d) in BH₄Ds.

**Initial biochemical presentation.** Diagnostically relevant and disease-specific constellations of biochemical parameters are presented in Fig. 6.

Along with typical changes in biogenic amines (i.e. reduced homovanillic acid (HVA) and 5-hydroxyindolacetic acid (5-HIAA), elevated 3-O-methyl-Dopa (3-OMD), levodopa (L-Dopa) and 5-hydroxytryptophan (5-HTP)), abnormalities of tetrahydrobiopterin

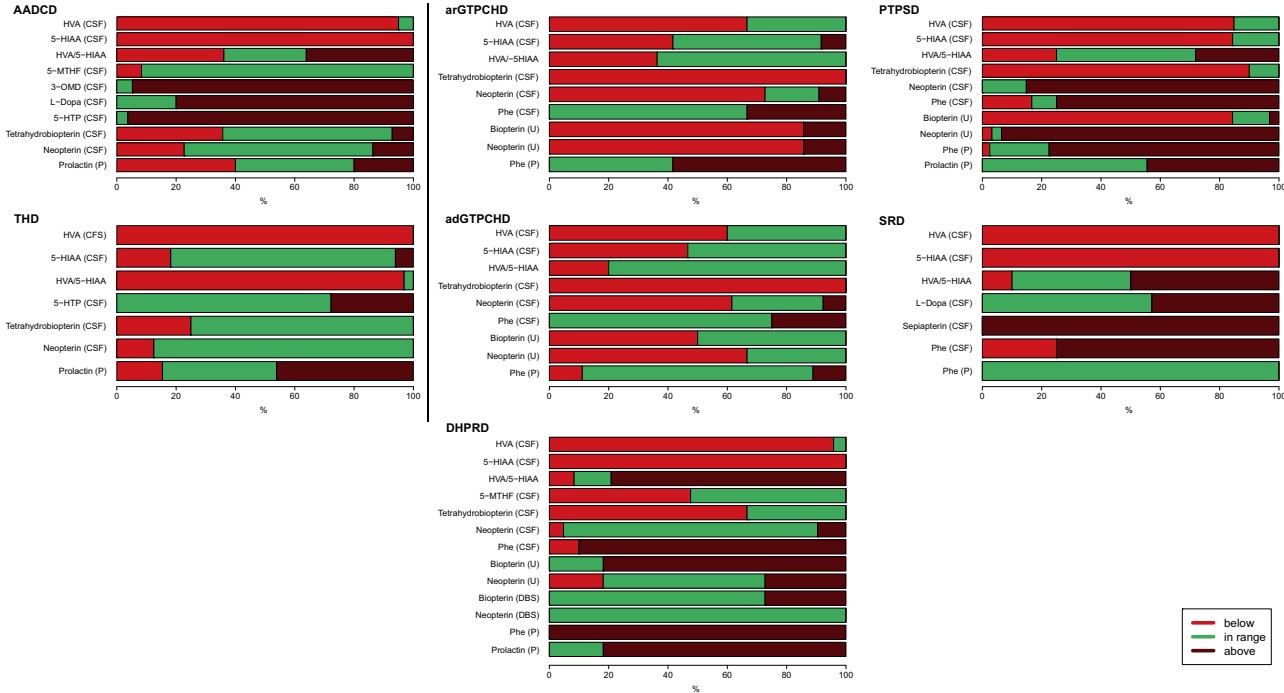

**Fig. 6 Biochemical pattern.** Disease-specific changes of biochemical parameters in plasma (P), dried blood spot (DBS), urine (U) and cerebrospinal fluid (CSF) before treatment.

and neopterin in CSF were observed in some AADCD patients (Fig. 6) Urinary vanillactic acid was not reported as an initial diagnostic parameter in our AADCD cohort. In THD, HVA and the ratio HVA/5-HIAA were decreased in almost all samples, while 5-HTP and 5-HIAA were typically normal. Prolactin in plasma showed high variability in both diseases (Fig. 6).

Phenylalanine (Phe) was normal or high in CSF and plasma in arGTPCHD while it was predominantly normal in both blood and CSF in adGTPCHD (Fig. 6). Phe was high in both CSF and in plasma in PTPSD and DHPRD. In SRD, Phe was increased in CSF while being normal in plasma, in line with previous reports[12]. Pterin disturbances were reported in CSF and in urine for adGTPCHD. Results on pterins in dried blood spots (DBS) in adGTPCHD were not available. 7-8-dihydrobiopterin in CSF was determined rarely but three measurements in DHPRD and two in SRD were high while being always normal in other BH4 disorders. HVA and 5-HIAA were more frequently decreased in PTPSD, DHPRD and SRD than in ad/arGTPCHD. In PTPSD and DHPRD prolactin was elevated in 44% and 82% of cases, respectively, while it was normal in almost all other BH4Ds. Decreased 5-methyltetrahydrofolate in CSF was reported only in DHPRD, except in one case with arGTPCHD and three cases with PTPSD.

**Fatal outcomes**. Three patients with AADCD and one with THD died during the study period. Death occurred at 2.4, 2.6 and 19.8 years of life in the AADCD patients. One patient died of pneumonia while in the remaining two cases the cause was unknown. The THD patient died at 13 years of age because of an acute lower respiratory tract infection.

## Discussion
The evaluation of 275 patients with disorders of biogenic amines (224 new and 51 previously published cases) that were analyzed using a standardized longitudinal approach revealed new phenotypic aspects of the initial clinical and biochemical

presentation, peri- and postnatal courses as well as diagnostic work-up. We present an increased incidence of prematurity in AADCD and of SGA in THD and in PTPSD. Patients with PTPSD were also prone to sIUGR and congenital microcephaly. We report one patient with DHPRD without HPA on NBS. We confirm the significant impact of HPA detection on NBS on the diagnostic work-up in a group of BH4Ds. Furthermore, we present the association of specific symptoms, such as oculogyric crises, dystonia, sleep and thermoregulation disorders, with age at diagnosis and diagnostic delay.

Pregnancies in both main disease groups were rarely complicated by medical problems. The issues described were most likely due to the pregnancy itself and not to the fetal disease. Exceptions were those cases in which mothers were affected by adGTPCHD, consistent with previous literature[13]. First symptoms of disorders of biogenic amines typically occur in the neonatal period or in infancy. Our data on the anthropometrical values at birth raise questions about prenatal disease manifestation. AADCD (18%) and SRD (21%) showed an increased rate of prematurity in our study compared to the global incidence of prematurity that is estimated as 9.6% ranging from 6.2% in Europe to 9.1 % in Asia, 10.6 % in North America and 11.9% in Africa[14]. Various causal factors such as fetal or maternal health conditions along with genetic, environmental, behavioral and socioeconomic factors as well as the differences in availability of preventive interventions between developed and developing countries influence the estimated rates. Since most of the patients in this study were born in Europe, North America or Asia, the background preterm birth rate may be expected to be between 6.2% and 10.6%. While the rate in SRD should be critically interpreted due the small number of cases, our data document an increased rate of prematurity in AADCD.

Additionally, the risk of SGA at birth is higher in PTPSD and THD (56% and 49%, respectively) compared to the considerable variation in the prevalence of infants born SGA, ranging from 4.6–15.3 % across Europe and 5.3% in east Asia to 41.5% in south Asia[15,16]. AADCD, arGTPCHD, DHPRD and SRD also show

elevated but not as high SGA rates. While neonates with PTPSD are also prone to sIUGR and congenital microcephaly (24%), we could not observe any trend towards microcephaly in DHPRD in contrast to the reported 25% rate in a historical cohort[17]. The observation that PTPSD patients were at high risk for prematurity could not be confirmed in our cohort but the detection of SGA and sIUGR are in line with the previously reported tendency to have very low birth weight (BW < 1500 gram[5,18]). To date, there are no previous reports of relevant changes in birth metrics in newborns with THD.

In previous publications the clinical phenotype of THD was divided into type A and severe type B[19], with higher frequency of perinatal abnormalities, including prematurity in patients with type B. In our iNTD study group of THD, we could not find a clear difference that would justify a differentiation into two types of severity. Furthermore, THD patients did not show an increased frequency of prematurity compared to the overall incidence as mentioned above. Perinatal abnormalities, postnatal problems, achievement of gross motor milestones including walking without assistance and medication varied regardless of the age of initial symptoms. Therefore, we propose to abandon this classification.

These observed peri- and postnatal changes indicate that impairments of biogenic amine neurotransmitter metabolism and their effects on the fetus start during pregnancy. Embryonic lethality was reported in *TH*, *DβH* and homozygous *GCH1* knock-out mice previously[20–22]. In mice, a rescue until term is possible with L-Dopa, $BH_4$ or dihydroxyphenylserine supplementation indicating that noradrenaline, dopamine and $BH_4$ are essential for fetal development. Homozygous *PTPS* and *Spr* knock-out mice, and *GCH1* knock-in mice are born visibly normal, have growth retardation postnatally and die after 48 h to a few weeks of age[23–26]. The $Qdpr^{-/-}$ mice are indistinguishable from their wildtype littermates and show normal growth[27]. Furthermore, genetically rescued *PTPS* knock-out, *TH* knock-in and *AADC* knock-in murine models are born without obvious morphological abnormality and survive but show growth retardation[23,28,29]. These animal models demonstrate that $BH_4$ regulates catecholamine synthesis through altering TH protein levels and that the postnatally expected increase of dopamine and TH protein concentration in the brain is disrupted by $BH_4$ deficiency[23,25]. Maternal compensation of $BH_4$ and dopamine deficiency as well as postnatal rescue has been demonstrated to be possible but limited[20–22,24,25,30]. Further studies will be needed to elucidate the disease-related pre- and postnatal findings of our study and the effectiveness and limitations of maternal metabolic compensation.

Nonspecific symptoms, such as feeding problems and hyperbilirubinemia were the most common postnatal problems in our study. A small group of patients with arGTPCHD presented with tremor, jitteriness, irritability and some patients with AADCD showed temperature instability and/or hypoglycemia postnatally that could be interpreted as disease-related. Following the postnatal period, the clinical presentation varied between diseases, although the symptoms are caused by a similar pathophysiological mechanism with the disruption in dopaminergic and/or serotoninergic neurotransmission. We showed that while AADCD presented with a variety of non-motor and motor symptoms[31,32], THD had an initial clinical picture clearly dominated by motor symptoms in addition to developmental delay starting on average in early infancy. In the case of $BH_4$Ds our results are similar to previous reports[5,17,33,34]. Seizures were typical in DHPRD and PTPSD among $BH_4$Ds while sleep disorders were especially frequent in SRD. These observations cannot be explained solely by any evidence but some hypotheses can be generated based on previous reports on different pathways: (1) Decreased $BH_4$ concentrations and elevated levels of 7,8-dihydrobiopterin ($BH_2$) in DHPRD and SRD, in the latter together

with elevated sepiapterin, lead to disturbances in intracellular $BH_4$:$BH_2$ ratio that codetermines uncoupling of endothelial NOS, resulting in generation of oxygen radicals[35,36]. Although the impact of these perturbations on the clinical picture remains unclear, interactions between NO concentration and NOS activity and epileptic discharges[37,38] as well as sleep initiation and maintenance[39] have been postulated. (2) The secondary cerebral folate deficiency occurring frequently in DHPRD[40] may result in severe epileptic encephalopathy[41,42]. (3) A disturbed melatonin homeostasis, documented by low urinary sulphatoxymelatonin levels in patients with SRD could provide another pathophysiological link[43].

Our data demonstrate that if recognized correctly, some disease-related symptoms could raise clinical suspicion and facilitate prompt diagnosis. Sleep and thermoregulation disturbances were associated with an earlier age at diagnosis and shorter latency to diagnosis, especially in AADCD. Oculogyric crises were associated with a longer diagnostic delay, although they correlated with an earlier age at diagnosis. This observation implies that oculogyric crises occur early and that correct recognition of the symptom could potentially shorten the diagnostic delay. Developmental delay was associated with a later age at diagnosis and a longer diagnostic delay in primary disorders of biogenic amine metabolism in contrast to $BH_4$Ds. This could be explained on one hand by the fact that developmental delay is a rather nonspecific symptom and on the other hand that it could be recognized more easily and earlier by a physician or parents. We could furthermore show an association of dystonia with higher age at diagnosis and a longer latency to diagnosis, probably due to the broad spectrum of differential diagnoses of dystonia[44].

Initial biochemical profiles largely revealed disease-specific patterns but our data showed occasional atypical biochemical findings that underline the importance of a careful interpretation of pterins and biogenic amines together with the clinical picture. For example, some patients with AADCD had abnormal concentrations of tetrahydrobiopterin and neopterin in CSF, in contrast to previous literature that emphasized the importance of normal pterin findings in AADCD[32]. Therefore, one should be aware of potential secondary disturbances most likely without clinical significance. Our data confirm that arGTPCHD can present with or without HPA but we also found some exceptions to the rule about which disorders present with HPA. Our registry contains one patient with a new variant in the *QDPR* gene showing repeatedly normal phenylalanine levels in plasma but high levels in CSF. This observation needs to be studied with further functional analyses.

Furthermore, we observed that ad/arGTPCHD and PTPSD may also have increased or normal HVA and 5-HIAA in CSF in contrast to DHPRD and SRD, in which HVA and 5-HIAA were mainly decreased. Pterin profiles tend to be consistent and disease-specific in $BH_4$Ds, except in DHPRD. Also, in line with previous reports, we observe that pterin measurements in urine and CSF are more sensitive than in DBS or plasma[3,17].

Hyperprolactinemia is frequently used as a marker of central dopaminergic deficiency, but normal prolactin levels do not exclude a neurotransmitter disorder[45]. Our data support this statement, as prolactin levels were scattered from low to high in disorders where cerebral dopaminergic deficiency with decreased HVA was documented. When interpreting blood prolactin, other causes for hyperprolactinemia must also be considered[46].

Our results also provide additional insights into the interdependence of disease-specific diagnostic delay and diagnostic tools. The suspected diagnoses were confirmed most frequently by mutational analysis alone or in combination with specific biochemical tests. Those diseases that had abnormal NBS due to HPA had a significant shorter diagnostic delay. Overall, DHPRD

could be diagnosed fastest, probably due to easily accessible DHPR activity measurement in DBS. In arGTPCHD and DHPRD a tendency to diagnose still asymptomatic children before disease onset, following a pathological NBS result was found. This underlines the importance of a complete, systematic and timely diagnostic work-up of every HPA as recommended in the recently published guidelines[17]. It is important to note, since arGTPCHD and DHPRD rarely may show normal phenylalanine concentrations on NBS, a diagnostic re-evaluation should be performed if there is strong clinical suspicion.

As adGTPCHD and SRD are not detectable on NBS for PKU, the finding of a prolonged diagnostic delay in these disorders is not unexpected. In our study, SRD had the highest latency to diagnosis as reported previously[5,33]. This is presumably caused by the absence of an easily accessible biochemical marker and the challenging clinical picture of SRD. In addition, characteristics of SRD such as oculogyric crises and sleep disorders may remain undetected or be misinterpreted. Complementary to the literature, our study demonstrated that patients with adGTPCHD may show symptoms as early as 6 months of age, and that the mean latency to diagnosis is approximately 5 years, shorter than the previously reported latency of 13–16 years[34,47]. Considering that PTPSD was first described in the 1980s[48], AADCD and THD in the 1990s[49,50] and SRD in 2001[51], our observation of a tendency to a shortened diagnostic delay in correlation with more recent birth dates confirms that enhanced knowledge and awareness of these diseases, efforts to establish a standardized algorithm for diagnostic work-up, and broadened availability of (genetic) diagnostic tools all likely reduce patients' diagnostic odyssey. The latter aspect can be supported by the fact that the implementation of next-generation sequencing-based diagnostics using approaches such as targeted resequencing, whole-exome and whole-genome sequencing has strongly changed clinical genetics[52,53]. Since the first whole-exome sequencing proof of concept experiments in 2009[54,55], the discovery of disease-causing genes using these technologies has increased rapidly. The impact of modern genetic diagnostic tools on the length of the diagnostic process in patients suspected of having rare genetic conditions has been[52,56] and will be an interesting subject for further studies.

In our study no patient with BH$_4$D was reported as deceased, in contrast to the mortality rate in PTPSD and DHPRD in a previous retrospective study[5]. This can be explained by the size of our study population and by the fact that only 10 out of 161 patients with BH$_4$Ds were born before the establishment of newborn screening programs in the 1980s.

Based on the fact that the inclusion criteria and requirements for patient enrollment into the iNTD registry were strictly defined and bound to complex ethical standards, the number of patients enrolled is not comparable with earlier retrospective registry studies (e.g. JAKE; http://www.biopku.org/home/jake.asp, BIODEF[57]; http:// www.biopku.org). Our registry provides data depending on the participating centers and does not present the overall incidence of the diseases worldwide. In addition, the medical history module of the registry analyzed in our study is the only part, which gathers data summarized by treating physicians, parents, caregivers as well as patients. Therefore, some information was retrospective or missing.

Following our observation of a high incidence of prematurity and SGA in some diseases, a revision of the exclusion criterion "BW < 1500 gram" will be proposed to the iNTD steering committee to avoid an underestimation of patients who are born preterm or with very low birth weight. The results of our study reinforce that international cooperation and patient registries are essential for a better understanding of rare diseases as well as for harmonization of diagnostic algorithms and standards of patient care in inherited disorders of neurotransmitter metabolism.

In conclusion, we describe comprehensive insights into pre-, peri- and postnatal presentations of inherited disorders of biogenic amines, as well as specific clinical and biochemical patterns affecting the diagnostic process. Our results emphasize the importance of recognizing the potential early signs and of careful and systematic clinical evaluation to improve diagnostic approaches in these rare neurodevelopmental diseases. These observations should serve as a basis for further studies on the evolving phenotypic spectrum in disorders of biogenic amines.

## Methods

**The International Working Group on Neurotransmitter related Disorders (iNTD).** The iNTD patient registry, which is web-based and password-protected (https://www.intd-registry.org), was approved by the Institutional Research Ethics Board (IRB) Heidelberg University Hospital (S-471/2014, registered German Clinical Trials Register, https://www.drks.de, DRKS00007878) on December 22$^{nd}$ 2014 and subsequently by all contributing centers[11]. All procedures followed were in accordance with the Helsinki Declaration of 1975, as revised in 2013. Written informed consent was obtained from all study participants or their legal guardians. iNTD was founded without any industry involvement or sponsorship.

Exclusion criteria of the iNTD patient registry: Patients with severe comorbidities, e.g. Down syndrome, intraventricular hemorrhage (°III-IV) in the newborn period, very low birth weight (<1500 gram), kernicterus, embryofetal disease due to maternal alcohol or drug abuse.

**Definitions.** Gestational age was calculated based on completed weeks of gestation. Preterm pregnancy was defined according to the International Classification of Disease as delivery before 37 completed (<37 + 0) weeks of gestation[58,59]. Small for gestational age (SGA) was defined as birth weight (BW) below the 10$^{th}$ percentile. Microcephaly was defined as head circumference at birth (BHC) below the 3$^{rd}$ percentile[60]. Symmetrical intrauterine growth restriction (sIUGR) was used as only referring to parameters available postnatally and was defined as BW, BHC and birth length (BL) below 10$^{th}$ percentile[61]. We defined the neonatal period as the first 30 days of life. Infancy referred to 31 days to 24 months, childhood to 3–12 years, adolescence 13–18 years and adulthood older than 18 years.

Initial symptoms are the first clinical findings that are considered to be disease related, observed by the physician, parents or the patient and can be both objective and subjective. The evaluation of the initial symptoms was based on retrospective data. The following symptoms were available in the medical history form in a controlled vocabulary: encephalopathy, developmental delay (psychomotor retardation), microcephaly, seizures, muscular symptoms, dystonia, dyskinesia, oculogyric crisis, thermoregulation disorders, sleep disorders and hypoglycemic episodes. Localization and type of muscular symptoms as well as type, frequency and length of seizures could be specified. Free text boxes were available for additional symptoms. The frequency of initial symptoms was analyzed in both age-dependent and age-independent manner. Additional symptoms included only in free text boxes were described age-independent.

Selective screening stands for a diagnostic process following onset of clinical symptoms. High-risk family screening describes a targeted diagnostic work-up initiated due to a confirmed disease case in the family, before or after onset of an individual's clinical symptoms.

To describe the initial biochemical presentation (at time of diagnosis), we analyzed the most frequently measured parameters among 30 biochemical parameters listed in the registry.

**Statistical analysis.** Statistical analyses were performed in R (version 4.0.2). Numeric variables were compared between two independent groups with Wilcox-Mann-Whitney (WMW) test, or $t$-test with Welch correction, setting $P < 0.05$ as significant. Analysis of variance (ANOVA) was used to compare numeric variables between more than two groups. The Benjamini-Hochberg adjustment was applied to correct type 1 error, when applicable. No a-priori hypotheses were tested and therefore all p-values had descriptive character. Classification and regression trees (CART) were used to identify constellations of symptoms that might have an impact on age at diagnosis or diagnostic delay (= age at diagnosis - age at initial symptom) for different diseases. Standard deviation scores (SDSs) for anthropometric variables at birth were computed according Fenton et al.[62]

<br>

**Reporting summary.** Further information on research design is available in the Nature Research Reporting Summary linked to this article.

## Data availability

All data supporting the findings described in this manuscript are not publicly available due to existing data protection laws but are available from the corresponding author (T.O.) upon reasonable request and within the limitations of the informed consent. All requests for raw and analyzed data will be reviewed by the iNTD executive board and iNTD members within 72 h. Data ownership is maintained by the members of the iNTD

network. All participating iNTD members approved this study. Source data for all the figures, tables 2–4 and supplementary table 1 are provided with the paper. Data transferred from the JAKE database (http://www.biopku.org/home/jake.asp) was not analyzed in this study. Source data are provided with this paper.

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

## Acknowledgements

We thank all patients and their families for their contributions to this study and for their trust. T.H. and J.K. were supported the grant from the Ministry of Health of the Czech Republic RVO-VFN 64165 GJIH-0599-00-7-846 and ProgresQ26/LF1. A.G.C. and N.J.P. are supported by FIS P118/00111 "Instituto de Salud Carlos III (ISCIII)" and "Fondo Europeo de desarrollo regional (FEDER)". T.O., K.J., G.F.H. and O.K.H. were supported in parts by the Dietmar Hopp Foundation, St. Leon-Rot, Germany. M.A.K. is funded by an NIHR Professorship, the Sir Jules Thorn Award for Biomedical Research and the Rosetrees trust. M.V. is supported by Stichting Stofwisselkracht Grant. D.H. acknowledges funding by the Molecular Diagnostics Program of the National Center for Tumor Diseases (NCT) Heidelberg. We are grateful for fruitful collaboration with the following clinical partners and patient support groups: Dr. Manolis Bilanakis from the University of Athens, Aghia Sofia Hospital, Athens, Greece, Prof. Carla Carducci, Dr. Claudia Carducci and Prof. Antonio Angeloni from the Department of Experimental Medicine, Sapienza University of Rome, Italy, DeNeu and Proyecto Pol, Spain, AADC Research Trust, United Kingdom and The association Lil' Brave One (Hrabriša), Serbia. We also thank the Society for the Relief of Disabled Children in Hong Kong for donations to support the commencement of CSF neurotransmitter analyses in Hong Kong.

## Author contributions

O.K.H. and T.O. conceived and designed the study. O.K.H., D.H. and S.F.G. preformed data analysis. O.K.H. drafted the initial manuscript. O.K.H, S.F.G. and D.H. prepared tables and figures. O.K.H., G.H., T.S.P., S.F.G. and T.O. contributed to finalize manuscript draft with input from all authors. O.K.H., G.H., E.C.S., Y.Y., M.M., R.P., J.F., S.M.A., S.W., T.S.P., D.I.Z., J.K., M.A.K., E.L.L., M.O., S.K., T.W., H.G., S.S.B., F.P., T.H., R.S., A.B., H. S.S., V.L., G.F.H., K.J., A.G.C., C.T.O., B.A., C.W.F., P.G., S.T.K.H., D.K., M.K., I.K., J.A.K., A.K., T.L., F.M., P.M., C.M., H.N.Ö.M., N.A.J.P., J.A.F.R., D.S., G.S., J.S.C., M.M.V. contributed to data and patient enrollment. All authors approved the final version of the manuscript.

## Funding

## Competing interests

A.G.C. receives teaching honorarium from PTC Therapeutics GT, Inc. C.M. and E.L.L. have received consultancy honorarium as part of the Advisory Board of PTC Therapeutics GT, Inc. G.F.H. receives teaching as well as consultancy honorarium from PTC Therapeutics GT, Inc. J.F. had trials with Biogen (Angelman's Syndrome) and Stealth Biotherapeutics (Mitochondrial Disorders); J.F.'s spouse is Founder and Principal of Friedman Bioventure, which holds a variety of publicly traded and private biotechnology interests. O.K.H. and T.W. have received teaching honorarium from PTC Therapeutics GT, Inc. R.P. has received honoraria as a speaker from Genesis Pharma, PTC Therapeutics GT, Inc. and as consultant in Advisory board of PTC Therapeutics GT, Inc. T.O. receives teaching honorarium and research support from PTC Therapeutics GT, Inc. V.L. has received consultancy honoraria as part of Advisory Boards organized by PTC Therapeutics GT, Inc., BioMarin Pharmaceutical Inc. and Homology Medicines. The remaining authors declare no competing interests.

## Additional information

[1]University Children's Hospital Heidelberg, Division of Child Neurology and Metabolic Disorders, Heidelberg, Germany. [2]University of British Columbia, Department of Pediatrics, Division of Biochemical Genetics, BC Children's Hospital, Vancouver, BC, Canada. [3]Inborn errors of metabolism Unit, Department of Neurology, Institut de Recerca Sant Joan de Déu and CIBERER-ISCIII, Barcelona, Spain. [4]Inborn Errors of Metabolism and Child Neurology Unit, Department of Pediatrics, Hospital Germans Trias i Pujol, Badalona and Faculty of Medicine, Universitat Autònoma de Barcelona, Barcelona, Spain. [5]Hacettepe University, Faculty of Medicine, Department of Pediatrics, Section of Metabolism, Ankara, Turkey. [6]Department of Human Neuroscience, Unit of Child Neurology and Psychiatry, Università degli Studi di Roma La Sapienza, Rome, Italy. [7]First Department of Pediatrics of the University of Athens, Aghia Sofia Hospital, Athens, Greece. [8]UCSD Departments of Neuroscience and Pediatrics; Rady Children's Hospital Division of Neurology, Rady Children's Institute for Genomic Medicine, San Diego, CA, USA. [9]Division of Clinical and Metabolic Genetics, Department of Pediatrics, University of Toronto, The Hospital for Sick Children 555 University Avenue Toronto, Toronto, ON, Canada. [10]Department of Pediatrics and Adolescent Medicine, The Hong Kong Childrenś Hospital, Hong Kong, Hong Kong. [11]Department of Neurology, Washington University School of Medicine, St. Louis, MO, USA. [12]First Department of Pediatrics Aristotle University of Thessaloniki Egnatia St. 106, Thessaloniki, Greece. [13]Department of Pediatrics and Inherited Metabolic Disorders, First Faculty of Medicine, Charles University and General University Hospital in Prague, Prague, Czech Republic. [14]Developmental Neurosciences, UCL Great Ormond Street-Institute of Child Health and Department of Neurology, Great Ormond Street Hospital, London, UK. [15]Pediatric Neurology Unit, Department of Pediatrics, University Hospital Reina Sofía, IMIBIC and CIBERER, Córdoba, Spain. [16]Childrenś Department Division of Child Neurology Oslo University Hospital Rikshospitalet Pb 4956 Nydalen, Oslo, Norway. [17]Çukurova University, Faculty of Medicine, Department of Pediatrics, Division of Pediatric Metabolism and Nutrition, Adana, Turkey. [18]Department of Neurology, Donders Institute for Brain, Cognition and Behaviour, Radboud University Medical Center, Nijmegen, The Netherlands. [19]Department of Pediatrics, Pediatric Neurology Unit, UZ Brussel VUB, Brussels, Belgium. [20]Department of Pediatrics, University of Alberta Glenrose Rehabilitation Hospital, Edmonton, AB, Canada. [21]Clinic for Pediatrics I, Medical University of Innsbruck, Innsbruck, Austria. [22]Department of Pediatrics, AOU Città della Salute e della Scienza, Torino, Italy.

[23]Children's Hospital, University Medical Center Hamburg-Eppendorf, Hamburg, Germany. [24]U.O.C. Malattie Metaboliche Ereditarie, Dipartimento della Salute della Donna e del Bambino, Azienda Ospedaliera Universitaria di Padova – Campus Biomedico Pietro d'Abano, Padova, Italy. [25]German Cancer Consortium (DKTK), Heidelberg, Germany. [26]Computational Oncology, Molecular Diagnostics Program, National Center for Tumor Diseases, DKFZ, Heidelberg, Germany. [27]Heidelberg Institute for Stem cell Technology and Experimental Medicine (HI-STEM), Heidelberg, Germany. [28]Department of Pediatric Immunology, Hematology and Oncology, Heidelberg University Hospital, Heidelberg, Germany. [29]University Children's Hospital Heidelberg, Dietmar-Hopp Metabolic Center, Heidelberg, Germany. *A list of authors and their affiliations appears at the end of the paper. ✉email: Thomas.Opladen@med.uni-heidelberg.de

## iNTD Registry Study Group

Birgit Assmann[1], Cheuk-Wing Fung[10], Philipp Guder[23], Stacey Tay Kiat Hong[30], Daniela Karall[21], Mitsuhiro Kato[31], Ivana Kavecan[32], Jeanette Aimee Koht[33], Alice Kuster[34], Thomas Lücke[35], Filippo Manti[6], Pablo Mir[36], Chris Mühlhausen[37], Halise Neslihan Önenli Mungan[17], Natalia Alexandra Julia Palacios[3], Joaquín Alejandro Fernández Ramos[15], Dora Steel[14], Galina Stevanović[38], Jolanta Sykut-Cegielska[39] & Marcel M. Verbeek[40,41]

[30]KTP-National University Children's Medical Institute, National University Health System, Singapore, Singapore. [31]Department of Pediatrics, Showa University School of Medicine, Tokyo, Japan. [32]Faculty of Medicine, University of Novi Sad, Institute for Children and Youth Health Care of Vojvodina, Novi Sad, Serbia. [33]Department of Neurology, Oslo University Hospital, Oslo, Norway. [34]Department of Neurometabolism and Metabolic Disorders, University Hospital of Nantes, Nantes, France. [35]University Children's Hospital, St. Josef-Hospital, Ruhr-University Bochum, Bochum, Germany. [36]Unidad de Trastornos del Movimiento Servicio de Neurología y Neurofisiología Clínica Unidad de Gestión Clínica de Neurociencias Instituto de Biomedicina de Sevilla (IBiS), Hospital Universitario Virgen del Rocío, Sevilla, Spain. [37]Department of Pediatrics and Adolescent Medicine, University Medical Centre Göttingen, Göttingen, Germany. [38]Clinic of Neurology and Psychiatry for Children and Youth, School of Medicine, University of Belgrade, Belgrade, Serbia. [39]Department of Inborn Errors of Metabolism and Pediatrics, Institute of Mother and Child, Warsaw, Poland. [40]Department of Neurology, Donders Institute for Brain, Cognition and Behavior, Radboud University Medical Center, Nijmegen, The Netherlands. [41]Department of Laboratory Medicine, Translational Metabolic Laboratory (TML), Radboud University Medical Center, Nijmegen, The Netherlands.

