## [Peer Review File · Nature Communications]

Reviewers' Comments:

Reviewer #1:

Remarks to the Author:

The manuscript collects clinical and biochemical data on 275 patients (of which 224 were unpublished) with inherited disorders of neurotransmitter metabolism, from 42 different centers from 26 countries, collected in an international registry. I think the manuscript is interesting for clinicians who work with neurometabolic diseases although it is quite difficult to read because of lots of abbreviations and numbers. To make it clearer and more informative to the clinicians who are not experts on neurotransmitter diseases, especially the figures need revision. I would suggest the following remarks:

1. This study included 275 patients. From how many families do they originate? That would be interesting to know especially in those diseases where the number of patients is very small.
2. Do all patients included in this study have genetic confirmation of their disease? It should be mentioned in the text.
3. The way Figure 1 is presented is somewhat misleading, because it shows different diseases in almost equal sized figure despite their very different numbers of cases (f.ex. AADC n=56 and DATD n=5). This is also true with Figure 4a.
4. In Figure 2 the symptoms should be in the same order in a) and b) to make it more readable and informative.
5. Figure 3 would be easier to read and to compare diseases if all charts would have all symptoms mentioned (y-axis) even though not all diseases present with all symptoms. The order of charts is unclear, it would be better to have THD next to AADC, and BH4-deficiencies in the second row.
6. Figure 6 needs clarification. The biochemical parameters (y-axis) are very difficult to read, because they are written with too small font and because they vary in every chart. I would suggest the authors to present a table with abnormalities in biochemical parameters and diseases associated with them.
7. The authors could discuss the role of next-generation sequencing methods (exome/genome sequencing, gene panels) in the nowadays diagnostics of these diseases.
8. High-risk family screening and selective screening should be explained.
9. Table 2 could be in the supplementary data.
10. Line 443 "Oculogyric" should probably be "Oculogyric crises".

Reviewer #2:

Remarks to the Author:

The manuscript is a well-written report of the clinical and laboratory findings of 275 patients enrolled in an international database known as the iNTD for international working group on neurotransmitter related disorders. There were some unexpected findings, such as an increased rate of prematurity, SGA, and congenital microcephaly (against historical population controls) in some of the disorders. The findings confirm features that have been described in these disorders, with some intuitive findings such as earlier diagnosis is associated with the presence of newborn screening or the presence of early peripheral hyperphenylalaninemia.

The study represents the largest cohort reported to date of standardized evaluations of neurotransmitter diseases in a longitudinal manner. The overall findings are that deep phenotyping requires multi-site cooperation with international collaborations in the case of ultra-rare disorders. The lack of finding a clear difference between the previously stratified forms of THD into type A and B is a helpful insight and the recommendation to abandon this classification is helpful.

A few issues for the authors to address:

1. The latency to diagnosis had some interesting findings. What may have accounted for the faster diagnoses after 2009 for AADC deficiency and after 2005 for tyrosine hydroxylase deficiency? Neither of these have peripheral hyperphenylalaninemia.
2. There are some observations that merit at least speculation as to an explanation:
 - a) Why are seizures more typical in DHPRD and PTPSD among BH4 disorders?
 - b) Why are sleep disorders more frequent in SRD?
 - c) Why was 3-OMD in CSF elevated in untreated PTPS and DPR deficiencies without an increase in

L-dopa or 5-HTP? This is difficult to understand.

REVIEWER COMMENTS

Reviewer #1 (Remarks to the Author):

The manuscript collects clinical and biochemical data on 275 patients (of which 224 were unpublished) with inherited disorders of neurotransmitter metabolism, from 42 different centers from 26 countries, collected in an international registry. I think the manuscript is interesting for clinicians who work with neurometabolic diseases although it is quite difficult to read because of lots of abbreviations and numbers.

- We agree that there are many abbreviations which is due to the high number of diseases presented in the manuscript. Since these diseases are comparable with regard to the biochemical and the pathophysiological background we had decided to present all three disease groups in one manuscript to enable a clear and helpful comparison. Unfortunately, this approach leads to a large number of abbreviations. We added a table presenting the classification used in the manuscript (Table 1) to facilitate the overview for the readers.

To make it clearer and more informative to the clinicians who are not experts on neurotransmitter diseases, especially the figures need revision. I would suggest the following remarks:

1. This study included 275 patients. From how many families do they originate? That would be interesting to know especially in those diseases where the number of patients is very small.

- Thank you for this advice. We have added the information about the total number of families in the text of the results section as follows: "The remaining cohort of patients with biogenic amine disorders consisted of 275 patients from 248 families (157 (57%) female,.....". In addition, we added this information in the column "Total number of patients (families)" for each disorder in Table 2.

2. Do all patients included in this study have genetic confirmation of their disease? It should be mentioned in the text.

- The vast majority of the patients had genetic confirmation. In the remaining cases, either enzymatic analysis or the classical pattern in the analysis of biogenic amines and pterins in CSF led to the diagnosis. We have added this information to the text as "All diagnoses were confirmed either by mutational analysis alone or by a combination of specific biochemical tests in CSF, urine and blood (Table 3). " and all the available numbers of confirmatory diagnostic work-up methods in Table 3 under the column "Confirmatory methods (number of available results)"

3. The way Figure 1 is presented is somewhat misleading, because it shows different diseases in almost equal sized figure despite their very different numbers of cases (f.ex. AADCD n=56 and DATD n=5). This is also true with Figure 4a.

- We agree with reviewer's remark. We added the number of patients to Fig 1 and Fig 4a.

4. In Figure 2 the symptoms should be in the same order in a) and b) to make it more readable and informative.

- We thank the reviewer for this suggestion but we would like to present the symptoms in descending order according to the frequency of occurrence to allow a better comparison of the two disease groups. In our opinion, this way the reader has a better and faster overview of the "typical and most common" initial symptoms per disease group.

5. Figure 3 would be easier to read and to compare diseases if all charts would have all symptoms mentioned (y-axis) even though not all diseases present with all symptoms.

- Figure 3 presents the most common initial symptoms and their age dependence. It is important to note that we have only presented symptoms reported in at least 5 patients in each disease (except for arGTPCHD, DHPRD, and SRD in at least 3 patients). Mentioning all symptoms, including those reported in less than 3 patients in each disease, would make this figure very complicated and confusing. To make the diseases in this figure easier to compare, we have aligned the order of symptoms and replaced the legend. The results of the age-independent analysis are described in the main text.

The order of charts is unclear, it would be better to have THD next to AADCDD, and BH4-deficiencies in the second row.

- It was our strategy within the manuscript to apply a recognizable pattern throughout the figures. For this purpose, we always presented disorders of biogenic amines on the left side and BH₄ disorders on the right side of the figure. To follow the reviewer's suggestion for better readability, we inserted a dividing line between parts a. and b. in Figure 3 and 6.

6. Figure 6 needs clarification. The biochemical parameters (y-axis) are very difficult to read, because they are written with too small font and because they vary in every chart. I would suggest the authors to present a table with abnormalities in biochemical parameters and diseases associated with them.

- Thank you for this important note. We made the following changes to improve the clarity of the figure: The order of the biochemical parameters has been changed. Since the same CSF parameters were measured in all diseases, we have now placed them on top of the graphs. The urine, DBS and plasma results, which differ within diseases, follow below. We have also changed the position of the DHPRD and SRD subfigures, replaced the legend, renamed the designation of the x-axis as " %" in all subfigures and added a dividing line between parts a. and b. We hope that these changes will contribute to a better understanding of the figure. Since numerical representations in tabular form already exist in various previous publications and we want to achieve a better overview of the frequency of elevated, reduced, or even normal laboratory values with this figure, we prefer to leave the figure with the described modifications.

7. The authors could discuss the role of next-generation sequencing methods (exome/genome sequencing, gene panels) in the nowadays diagnostics of these diseases.

- We thank the reviewers for addressing this very important issue. We have included the role of these technologies as well as their likely impact on the shortened diagnostic delay in patients with younger birth dates in the discussion as follows ".....and broadened availability of (genetic) diagnostic tools all likely reduce patients' diagnostic odyssey. The latter aspect can be supported by the fact that the implementation of next-generation sequencing-based diagnostics using approaches such as targeted resequencing, whole-exome and whole-genome sequencing has strongly changed the clinical genetics^{43,44}. Since the first whole-exome sequencing proof of concept experiments in 2009^{45,46}, the discovery of disease-causing genes using these technologies has increased rapidly. The impact of modern genetic diagnostic tools on the length of the diagnostic process in patients suspected of having rare genetic conditions has been^{43,47} and will be an interesting subject for further studies."

8. High-risk family screening and selective screening should be explained.

- We have added an explanation of these screening methods in “Methods” section as follows:

“Selective screening stands for a diagnostic process following onset of clinical symptoms. High-risk family screening describes a targeted diagnostic work-up initiated due to a confirmed disease case in the family, before or after onset of an individual’s clinical symptoms.”

9. Table 2 could be in the supplementary data.

- We moved Table 2 to the supplementary data and named it “Supplementary Table 1”.

10. Line 443 “Oculogyric” should probably be “Oculogyric crises”.

- Corrected

Reviewer #2 (Remarks to the Author):

The manuscript is a well-written report of the clinical and laboratory findings of 275 patients enrolled in an international database known as the iNTD for international working group on neurotransmitter related disorders. There were some unexpected findings, such as an increased rate of prematurity, SGA, and congenital microcephaly (against historical population controls) in some of the disorders. The findings confirm features that have been described in these disorders, with some intuitive findings such as earlier diagnosis is associated with the presence of newborn screening or the presence of early peripheral hyperphenylalaninemia.

The study represents the largest cohort reported to date of standardized evaluations of neurotransmitter diseases in a longitudinal manner. The overall findings are that deep phenotyping requires multi-site cooperation with international collaborations in the case of ultra-rare disorders.

The lack of finding a clear difference between the previously stratified forms of THD into type A and B is a helpful insight and the recommendation to abandon this classification is helpful.

A few issues for the authors to address:

1. The latency to diagnosis had some interesting findings. What may have accounted for the faster diagnoses after 2009 for AADC deficiency and after 2005 for tyrosine hydroxylase deficiency? Neither of these have peripheral hyperphenylalaninemia.

- We thank the reviewer for this question. We have included the role of these technologies as well as their likely impact on the shortened diagnostic delay in patients with younger birth dates in the manuscript as follows“and broadened availability of (genetic) diagnostic tools all likely reduce patients’ diagnostic odyssey. The latter aspect can be supported by the fact that the implementation of next-generation sequencing-based diagnostics using approaches such as targeted resequencing, whole-exome and whole-genome sequencing has strongly changed the clinical genetics^{43,44}. Since the first whole-exome sequencing proof of concept experiments in 2009^{45,46}, the discovery of disease-causing genes using these technologies has increased rapidly. The impact of modern genetic diagnostic tools on the length of the diagnostic process in patients suspected of having rare genetic conditions has been^{43,47} and will be an interesting subject for further studies.”

2. There are some observations that merit at least speculation as to an explanation:
 - a) Why are seizures more typical in DHPRD and PTPSD among BH4 disorders?
 - b) Why are sleep disorders more frequent in SRD?

We thank the reviewer for these questions. It is important to mention that for the questions raised, in fact, only speculations are possible. After detailed consultation with co-authors and a renewed literature search, we could generate the following hypotheses and have added those in the discussion part as follows: "These observations cannot be explained solely by any evidence but some hypotheses can be generated based on previous reports on different pathways: (1) Decreased BH4 concentrations and elevated levels of 7,8-dihydrobiopterin (BH2) in DHPRD and SRD, in the latter together with elevated sepiapterin, lead to disturbances in intracellular BH4:BH2 ratio that codetermines uncoupling of endothelial NOS, resulting in generation of oxygen radicals^{1,2}. Although the impact of these perturbations on the clinical picture remains unclear, interactions between NO concentration and NOS activity and epileptic discharges^{3,4} as well as sleep initiation and maintenance⁵ have been postulated. (2) The secondary cerebral folate deficiency occurring frequently in DHPRD⁶ may result in severe epileptic encephalopathy^{7,8}. (3) A disturbed melatonin homeostasis, documented by low urinary sulphatoxymelatonin levels in patients with SRD could provide another pathophysiological link⁹."

c) Why was 3-OMD in CSF elevated in untreated PTPS and DPR deficiencies without an increase in L-dopa or 5-HTP? This is difficult to understand.

- We have double-checked the data in the registry and confirmed that the entries were analyzed correctly. In the second step we have asked each treating physician in each center to re-confirm the data. We have found that in rare specific cases the values entered were not collected before the start of therapy as intended. Thus, the 3-OMD results were entered for cases with 2 PTPSD and 1 DHPRD under L-dopa/Carbidopa therapy and in 1 PTPSD after L-Dopa trial. In one other patient, the age at CSF sampling had to be corrected, resulting in a different reference range (1 PTPSD). In one patient (DHPRD), the increase was less than 10% (55 vs. 50 nmol/l), which is not relevant in this dimension. In the remaining 1 case we could not receive a feedback from the physician so we have decided to not report it. In conclusion, we have removed the parts in the result and corrected the discussion with regard to this observation. We have adjusted the Figure 6.

1. 1. Crabtree, M.J. & Channon, K.M. Synthesis and recycling of tetrahydrobiopterin in endothelial function and vascular disease. *Nitric oxide : biology and chemistry* **25**, 81-88 (2011).
2. 2. Gao, L., *et al.* Sepiapterin reductase regulation of endothelial tetrahydrobiopterin and nitric oxide bioavailability. **297**, H331-H339 (2009).
3. 3. Jayakumar, A.R., Sujatha, R., Paul, V., Puviarasan, K. & Jayakumar, R. Involvement of nitric oxide and nitric oxide synthase activity in anticonvulsive action. *Brain Res Bull* **48**, 387-394 (1999).
4. 4. Kovacs, R., *et al.* Endogenous nitric oxide is a key promoting factor for initiation of seizure-like events in hippocampal and entorhinal cortex slices. *J Neurosci* **29**, 8565-8577 (2009).

5. 5. Greene, R.W. Role for neuronal nitric oxide synthase in sleep homeostasis and arousal. **110**, 19982-19983 (2013).
6. 6. Xu, F., *et al.* Disturbed biopterin and folate metabolism in the Qdpr-deficient mouse. *FEBS Lett* **588**, 3924-3931 (2014).
7. 7. Pope, S., Artuch, R., Heales, S. & Rahman, S. Cerebral folate deficiency: Analytical tests and differential diagnosis. *Journal of Inherited Metabolic Disease* **42**, 655-672 (2019).
8. 8. Kuseyri Hübschmann, O., *et al.* Brain MR patterns in inherited disorders of monoamine neurotransmitters: An analysis of 70 patients. *J Inherit Metab Dis* (2021).
9. 9. Batllori, M., *et al.* Urinary sulphatoxymelatonin as a biomarker of serotonin status in biogenic amine-deficient patients. *Sci Rep* **7**, 14675-14675 (2017).

Reviewers' Comments:

Reviewer #1:

Remarks to the Author:

The authors have addressed all points. I have no further comments or criticism.

Reviewer #2:

Remarks to the Author:

Thank you for the edits made to the manuscript including the additional explanations and references on the issues of specific epilepsies and sleep disorders being worse in certain bioamine synthesis conditions.